# Development of a Method for Producing Recombinant Human Granulocyte-Macrophage Colony-Stimulating Factor Using Fusion Protein Technology

**DOI:** 10.3390/cimb47090681

**Published:** 2025-08-25

**Authors:** Ekaterina A. Volosnikova, Tatiana I. Esina, Natalia V. Volkova, Svetlana V. Belenkaya, Yana S. Gogina, Galina G. Shimina, Elena A. Vyazovaya, Svetlana G. Gamaley, Elena D. Danilenko, Dmitriy N. Shcherbakov

**Affiliations:** State Research Center of Virology and Biotechnology “Vector”, Rospotrebnadzor, 630559 Koltsovo, Russia; esina_ti@vector.nsc.ru (T.I.E.); volkova_nv@vector.nsc.ru (N.V.V.); belenkaya.sveta@gmail.com (S.V.B.); gogina_yas@vector.nsc.ru (Y.S.G.); shimina_gg@vector.nsc.ru (G.G.S.); vyazovaya_ea@vector.nsc.ru (E.A.V.); gamaley_sg@vector.nsc.ru (S.G.G.); danilenko_ed@vector.nsc.ru (E.D.D.); dnshcherbakov@gmail.com (D.N.S.)

**Keywords:** recombinant proteins, rhGM-CSF, SUMO fusion technology, protein solubility enhancement, therapeutic protein production, protein chromatography, hematopoietic activity

## Abstract

Granulocyte-macrophage colony-stimulating factor (GM-CSF) is a multifunctional cytokine with therapeutic applications in oncology and neurodegenerative diseases. However, its clinical use is limited by the high cost of eukaryotic production systems. Here, we developed a cost-effective *Escherichia coli*-based platform for high-yield production of biologically active recombinant human GM-CSF (rhGM-CSF) using SUMO fusion technology. The engineered pET-SUMO-GM plasmid enabled expression of a 33 kDa fusion protein, accounting for 23–25% of total cellular protein, though it primarily accumulated in inclusion bodies. A multi-step purification strategy—including nickel affinity chromatography, Ulp protease cleavage, and hydrophobic chromatography—yielded >99.5% pure rhGM-CSF. In vitro functional assays demonstrated equivalent activity to the WHO international standard (ED50: 0.045 vs. 0.043 ng/mL in TF-1 cell proliferation). In vivo, the preparation significantly restored neutrophil counts (3.4-fold increase, *p* ≤ 0.05) in a murine cyclophosphamide-induced myelosuppression model. Our results establish a scalable, prokaryotic-based method to produce functional rhGM-CSF, overcoming solubility and folding challenges while maintaining therapeutic efficacy. This approach could facilitate broader clinical and research applications of GM-CSF, particularly in resource-limited settings.

## 1. Introduction

Granulocyte-macrophage colony-stimulating factor (GM-CSF) is a multifunctional cytokine with unique therapeutic potential. Initially developed to treat chemotherapy-induced neutropenia [1], this protein exhibits remarkable functional diversity, finding applications in oncology and neurodegenerative disease therapy.

In cancer treatment, recombinant GM-CSF (Sargramostim, Leukine^®^) not only restores hematopoiesis [2] but also serves as an immunotherapy component by enhancing antitumor immune responses through dendritic cell and macrophage activation [3]. Concurrently, neurology research highlights its neuroprotective properties: clinical studies demonstrate GM-CSF’s ability to improve cognitive function in Alzheimer’s disease [4,5] and promote neuronal survival in Parkinson’s disease [5,6]. This dual therapeutic potential stems from GM-CSF’s immunomodulatory effects—boosting antitumor immunity in oncology while suppressing neuroinflammation and facilitating pathological protein clearance in neurology [5].

Human GM-CSF is a glycoprotein composed of 127 amino acid residues. Its structure is stabilized by two disulfide bonds formed by four cysteine residues, resulting in a compact globular fold featuring four α-helices connected by loop regions. In its extracellular form, GM-CSF exists as a homodimer with two N-glycosylation sites (Asn27 and Asn37) and three O-glycosylation sites (Ser7, Ser9, and Thr10). Depending on the degree of glycosylation, several GM-CSF variants can be distinguished: the most heavily glycosylated form (28–32 kDa) carries two N-linked carbohydrate moieties, a partially glycosylated variant (23–25 kDa) contains only one N-linked glycan, and a minimally glycosylated form (16–18 kDa) possesses a single O-linked carbohydrate modification [7].

Eukaryotic expression systems are generally preferred for GM-CSF production as they ensure proper protein folding and post-translational modifications. CHO cell-derived GM-CSF in particular demonstrates prolonged bioactivity with delayed clearance compared to non-glycosylated variants [7]. However, the development of more economical production systems remains crucial for broader therapeutic applications [3,8]. While *Escherichia coli* offers a cost-effective alternative, it presents challenges including low solubility and inclusion body formation [9]. Notably, carbohydrate removal—whether through bacterial expression or targeted mutagenesis—does not compromise GM-CSF’s specific activity [10]

Fusion protein technology, particularly SUMO (small ubiquitin-related modifier), has emerged as a leading strategy to enhance recombinant protein expression and simplify purification in *E. coli*. SUMO’s small size and the availability of highly specific SUMO proteases for tag removal make it an ideal fusion partner [11,12,13].

The objective of this study was to engineer a recombinant plasmid for expression of a human GM-CSF-SUMO fusion protein, develop an efficient method for producing biologically active GM-CSF, and characterize the functional properties of the purified protein.

## 2. Materials and Methods

### 2.1. Oligonucleotides and Genes

The oligonucleotides listed in Table 1 were chemically synthesized (Biosset, Novosibirsk region, Russia).

The nucleotide sequence encoding a human GM-CSF was obtained from the GenBank database (accession no. M11220), codon-optimized and synthesized in the pGH-GM plasmid (DNA-Synthesis, Moscow, Russia).

### 2.2. pET-SUMO-GM Plasmid Construction

Amplification of SUMO and GM nucleotide sequences was performed using two primer pairs: GSUMO-F/SGM-R and GM-SUMO-R/SGM-F. The pGH-GM plasmid served as a template for GM gene amplification, while the pET SUMO plasmid was used for SUMO amplification. The PCR products of SUMO and GM were mixed and subjected to overlap extension PCR using flanking primers. The resulting PCR product and pET21a(-) vector were digested with FauNDI and CciNI restriction enzymes (SibEnzyme, Novosibirsk, Russia). The digested fragments were then mixed and ligated using T4 bacteriophage DNA ligase (SibEnzyme, Novosibirsk, Russia) for 30 min at room temperature. Transformation of competent *E. coli* NEB Stable cells (New England Biolabs, Ipswich, MA, USA) with the ligation products was performed using the heat shock method. Insert presence was confirmed by restriction analysis and Sanger sequencing. After verifying successful insertion, *E. coli* BL21(DE3) cells were transformed with the pET-SUMO-GM plasmid.

### 2.3. Cultivation

*Escherichia coli* BL21(DE3) cells transformed with the recombinant pET-SUMO-GM expression plasmid were grown on LB agar (neoFroxx, Einhausen, Germany) supplemented with kanamycin 50 µg/mL (Forever Pharmacy, Nanjing, China) at 37 °C for 18–20 h. A loopful of cells was then transferred into Lennox LB broth (pH 7.2) containing kanamycin (50 µg/mL) and incubated with shaking under the same conditions (overnight culture). The overnight culture was subcultured (1:100 dilution) into fresh LB broth with kanamycin (50 µg/mL) and incubated at 37 °C with continuous shaking (180 rpm). Optical density (OD) was monitored at 600 nm. Target protein expression was induced by adding 0.1 mM isopropyl β-D-1-thiogalactopyranoside (IPTG) at an OD600 of 0.9–1.1, and cultivation continued until the early stationary phase. Cells were harvested by centrifugation at 8000 rpm (7500× *g*) for 15 min at 4 °C.

### 2.4. Analysis of Target Protein Content

Samples were analyzed by denaturing electrophoresis in 12% sodium dodecyl sulfate-polyacrylamide gel (SDS-PAGE), followed by Coomassie R-250 staining (AppliChem, Darmstadt, Germany). The target protein content in bacterial cells was quantified by gel densitometry using a GelDoc Go imaging system (Bio-Rad, Hercules, CA, USA) with Image Lab software.

Protein concentration was assessed using the Lowry method.

### 2.5. Extraction and Refolding of SUMO-GM Fusion Protein

The *E. coli* BL21(DE3)/pET-SUMO-GM biomass was suspended in lysis buffer (20 mM Tris-HCl (Sigma-Aldrich, Irvine, UK), 1 mM phenylmethylsulfonyl fluoride (GERBU, Gaiberg, Germany), pH 8.0) at a 1:10 (w/v) ratio for 40 min. Cell disruption was performed with constant cooling on ice using an ultrasonic homogenizer Sonicator Q2000 (Qsonica, Newtown, CT, USA) at 40% amplitude with 5-s pulses and 5-s intervals. Sonication was stopped when the optical density of the suspension at 595 nm decreased by 45–55% from the initial value (optical density prior to sonication). The inclusion body (IB) pellet was separated by centrifugation at 12,000 rpm for 40 min at 4 °C.

The inclusion bodies isolated from the lysate were sequentially washed to remove bacterial contaminants using three buffer systems: first with 20 mM Tris-HCl, 5 mM EDTA (neoFroxx, Einhausen, Germany), pH 8.0 (twice), then with 20 mM Tris-HCl, 0.5% Tergitol 15-S-9 (neoFroxx, Einhausen, Germany), pH 8.0 (twice), and finally with 20 mM Tris-HCl, pH 8.0 (once). During each washing step, the IB suspension was incubated for 20 min at 2–8 °C with constant agitation using an MMS-3000 magnetic stirrer (BioSan, Riga, Latvia), followed by centrifugation at 12,000 rpm for 30 min at 4 °C and subsequent resuspension in the respective washing buffer.

Recombinant protein extraction from the washed IBs was performed by solubilization in the presence of a chaotropic agent (20 mM Tris-HCl, 8 M urea (neoFroxx, Einhausen, Germany), pH 8.0), using a solubilization buffer volume equal to the lysis buffer volume. β-mercaptoethanol (AppliChem, Darmstadt, Germany) was added to the suspension at a concentration of 20 mM to reduce interchain disulfide bonds [14]. The denatured protein solution was incubated for 22 h at 2–8 °C, after which insoluble cellular debris was removed by centrifugation at 12,000 rpm for 20 min at 4 °C.

The refolding process was initiated by slow, dropwise addition of the denatured protein solution (in 20 mM Tris-HCl, 8 M urea, 20 mM β-mercaptoethanol, pH 8.0) into a 10-fold volume of pre-chilled (4 °C) refolding buffer (20 mM Tris-HCl, pH 8.0) under constant gentle stirring. This method achieved a rapid dilution of denaturant (final urea concentration ~0.8 M) and oxidizing conditions favorable for disulfide bond formation (due to the dilution of the reducing agent β-mercaptoethanol and access to atmospheric oxygen).

The refolding reaction was allowed to proceed for 22 h at 4 °C with continuous mild agitation. To remove insoluble aggregates formed during refolding, the solution was centrifuged at 12,000× *g* for 30 min at 4 °C. The supernatant containing the soluble, refolded protein was then carefully collected for subsequent purification steps.

### 2.6. Affinity Chromatography of SUMO-GM Protein

The refolded protein solution was loaded onto a column containing Ni^2+^-charged IMAC Seplife FF resin (Sunresin, Xi’an, China) pre-equilibrated with 20 mM imidazole, 20 mM Tris-HCl, pH 8.0 buffer (PanReac AppliChem ITW Reagents, Germany) using an ÄKTA pure 150 medium-pressure chromatography system with Unicorn 7.6 software (Cytiva, Uppsala, Sweden). After sample application, the resin was washed with three column volumes (3 CV) of the equilibration buffer. The SUMO-GM fusion protein was then eluted with 8 CV of 350 mM imidazole, 20 mM Tris-HCl, pH 8.0 buffer, followed by a final column wash with 2 CV of 1 M imidazole, 20 mM Tris-HCl, pH 8.0 buffer.

### 2.7. Ulp Protease-Mediated Cleavage of SUMO-GM Fusion Protein

The SUMO-GM fusion protein was digested with ubiquitin-like protease (Thermo Fisher Scientific, Waltham, MA, USA) at a 1:50 (w/w) enzyme-to-substrate ratio. The resulting hydrolysate was dialyzed against 20 mM Tris-HCl buffer (pH 8.0) to remove imidazole and sodium chloride (neoFroxx, Einhausen, Germany) from the protein solution.

### 2.8. Purification of Recombinant Human GM-CSF

The dialyzed hydrolysate was loaded onto a Ni^2+^-charged IMAC Seplife FF column (Sunresin, Xi’an, China) equilibrated with 20 mM Tris-HCl (pH 8.0). The SUMO tag was eluted with 3 column volumes (CV) of 20 mM Tris-HCl, 1 M imidazole (pH 8.0), while the recombinant human GM-CSF (rhGM-CSF) flowed through unbound. The rhGM-CSF-containing flow-through fractions were pooled and dialyzed against 2 M NaCl, 20 mM Tris-HCl (pH 8.0).

The pooled fractions were then applied to a Phenyl Seplife column equilibrated with 2 M NaCl, 20 mM Tris-HCl (pH 8.0). Under these conditions, the target protein did not bind and was collected in the flow-through. Fractions containing rhGM-CSF with >95% purity were pooled and dialyzed against 20 mM Tris-HCl, 150 mM NaCl (pH 8.0) at 2–8 °C for 18–22 h. The protein solution was sterilized by filtration through a 0.22 μm PES membrane syringe filter (Jet Biofil, Guangzhou, China) and stored at −20 °C.

### 2.9. Experimental Animals

The study was conducted on 17 male CBA/CaLac mice (age 2.0–2.5 months, body weight 20–24 g) obtained from the breeding facility of SRC VB “Vector” of Rospotrebnadzor. Following an adaptation quarantine period, the mice were maintained under standard vivarium conditions with controlled temperature and humidity, having free access to food and water at all times. All experimental procedures were performed in accordance with the International Convention for the Protection of Vertebrate Animals Used for Experimental and Scientific Purposes (Strasbourg, 1986) and EU Directive 2010/63/EU on the protection of animals used for scientific purposes. The study protocol (No. 2, dated 14 February 2024) was approved by the Bioethics Committee of SRC VB “Vector” of Rospotrebnadzor.

### 2.10. In Vivo Evaluation of GM-CSF Hemostimulatory Activity

This experiment utilized a myelosuppression model induced by single intraperitoneal administration of 200 mg/kg cyclophosphamide (Sigma-Aldrich, Irvine, UK) to CBA/CaLac mice. Animals were divided into two experimental groups (n = 6 each). Twenty-four hours post-cytostatic administration, the test group received daily subcutaneous injections of recombinant human GM-CSF (90 μg/kg in 0.2 mL per 20 g body weight) for 4 consecutive days, while the control group received equivalent volumes of physiological saline following the same regimen. The intact group received no injections. All procedures were consistently performed in the morning.

Following the treatment course, tail vein blood samples were collected for analysis. Total leukocyte counts, along with relative and absolute neutrophil counts and other leukocyte differentials, were determined using a Goryaev counting chamber and Mikmed-6 microscope (LOMO, Saint Petersburg, Russia).

Statistical analysis was performed using Statgraphics Plus version 5.1 software (Statistical Graphics Corp., Cary, NC, USA). Due to the small sample sizes, non-parametric statistical methods were employed, including the Mann–Whitney U test for two-group comparisons and the Kruskal–Wallis H test for multiple group comparisons, with a critical significance level set at *p* ≤ 0.05.

### 2.11. In Vitro Assessment of GM-CSF Biological Activity

The biological activity of rhGM-CSF samples was evaluated by measuring the stimulation of proliferation in cytokine-dependent human erythroleukemia TF-1 cells using the XTT assay. This method is based on the ability of metabolically active cells to reduce the tetrazolium salt XTT (2,3-bis(2-methoxy-4-nitro-5-sulfophenyl)-2H-tetrazolium-5-carboxanilide) to formazan in the presence of phenazine methosulfate (PMS). For activity calibration, we used the WHO International Standard for recombinant human GM-CSF (1st Standard for Granulocyte Macrophage Colony Stimulating Factor (Human, rDNA Derived), NIBSC code: 88/646, United Kingdom (https://nibsc.org/documents/ifu/88-646.pdf, accessed on 5 June 2025).

The analysis was performed according to the method described in the authors’ previous work [15]. Two-fold serial dilutions of both test and standard preparations were tested in the concentration range of 8 ng/mL to 0.031 ng/mL. TF-1 cell suspensions in RPMI medium supplemented with 10% fetal bovine serum (50 μL, 10^4^ cells/well) were incubated with rhGM-CSF samples (50 μL/well) for 72 h at 37 °C in a humidified atmosphere containing 5% CO_2_. Then, 50 μL of XTT/PMS reagent mixture (CDH) was added to each test well, followed by additional 3-h incubation under the same conditions. Optical density was measured at 490/620 nm using a Varioskan LUX microplate reader (Thermo Fisher Scientific, Waltham, MA, USA).

Each dilution of both test and standard samples was analyzed in quintuplicate. The mean values for different rhGM-CSF concentrations were used to calculate the proliferation activity index (percentage relative to untreated control cells, set as 100%). The ED50 value, representing the protein concentration required to achieve a 2-fold proliferative effect compared to the control, was determined from the proliferation index vs. concentration plot. Statistical analysis was performed using Statgraphics version 5.0 (Statistical Graphics Corp., USA).

## 3. Results

### 3.1. Expression of Fusion Protein SUMO-GM in E. coli

For production of recombinant human GM-CSF, we developed a pET-SUMO-GM plasmid construct based on the pET expression system with a T7 bacteriophage promoter. This system was selected due to its high efficiency and widespread use for recombinant protein production in *E. coli*.

Sanger sequencing analysis confirmed the correct plasmid construction (Figure 1A). The resulting plasmid contained a SUMO-GM fusion gene, where the SUMO protein was fused to the N-terminus of GM-CSF (Figure 1B). This design enabled high-level expression of the target protein in the bacterial expression system.

Electrophoretic analysis demonstrated that cultivation of the recombinant *E. coli* BL21(DE3)/pET-SUMO-GM strain in the presence of IPTG inducer for 4–5 h resulted in efficient accumulation of the SUMO-GM fusion protein with a molecular weight of 33–34 kDa, accounting for 23–25% of total cellular protein (Figure 2). The wet biomass yield reached 3.8 ± 0.2 g per liter of culture medium, indicating high productivity of the engineered producer strain.

The target protein predominantly accumulated in inclusion bodies, requiring the development of a multi-stage washing protocol for purification from cellular contaminants that included EDTA buffer treatment for nucleic acid removal, detergent washing to eliminate membrane proteins, and a final low-ionic-strength buffer wash, with electrophoretic analysis confirming the protocol’s efficiency by demonstrating near-complete retention of the target protein in the inclusion body fraction while effectively removing contaminating proteins in the wash supernatants (Figure 3).

The protein was solubilized in 8 M urea. To facilitate correct folding, the native structure was restored by gradual refolding via stepwise dilution of the denaturant.

### 3.2. Purification and Characterization of the Recombinant Protein

The SUMO-GM fusion protein was purified by Ni-IMAC (nickel immobilized metal affinity chromatography), leveraging the N-terminal His-tag on SUMO for efficient binding (Figure 4). Following Ulp protease treatment to specifically cleave the SUMO-rhGM-CSF junction, an additional Ni-IMAC step was performed to remove the SUMO tag fragment.

The insufficient purity of rhGM-CSF following affinity chromatography (Figure 5) necessitated an additional hydrophobic interaction chromatography (HIC) step. Using Phenyl Seplife resin effectively removed contaminating proteins that bound to the matrix, unlike rhGM-CSF, yielding a highly purified preparation with 99.5% monomer content.

### 3.3. Biological Activity Assessment

The biological activity of the purified preparation was evaluated both in vitro and in vivo. Specific activity of the rhGM-CSF preparation was assessed using cell culture assays and a murine myelosuppression model.

For in vitro analysis, biological activity was determined by measuring the proliferation of TF-1 cells stimulated with the test sample. The WHO International Standard (1st Standard for Granulocyte Macrophage Colony Stimulating Factor, Human, rDNA Derived; NIBSC code: 88/646) served as a reference, containing 1 μg recombinant human GM-CSF with an activity of 10,000 IU per ampoule.

Comparative analysis of proliferative activity (Figure 6) revealed nearly identical dose–response curves for both preparations. The calculated ED50 values (concentration yielding two-fold proliferation versus control) were 0.043 ng/mL for the reference standard and 0.045 ng/mL for our rhGM-CSF preparation, demonstrating equivalent biological potency to the international standard.

The hematopoietic activity of rhGM-CSF was confirmed in a cyclophosphamide-induced myelosuppression mouse model. Peripheral blood leukocyte and segmented neutrophil counts were measured relative to saline-treated controls (set as 100%). As presented in Table 1, administration of cyclophosphamide (200 mg/kg) predictably decreased both total leukocyte and segmented neutrophil counts. Notably, animals treated with rhGM-CSF showed statistically significant (*p* ≤ 0.05) increases in both relative (294%) and absolute (340%) segmented neutrophil counts compared to controls (Table 2).

The results demonstrate that the recombinant human GM-CSF preparation produced using fusion protein technology exhibits potent hematopoietic activity.

## 4. Discussion

GM-CSF holds significant therapeutic potential in oncology and neurology, but its broader clinical use has been limited by the high costs and complexity of eukaryotic production systems. In this study, we developed a cost-effective *E. coli*-based platform for high-yield production of biologically active recombinant human GM-CSF using SUMO fusion technology. Although the protein accumulated in inclusion bodies, the fusion tag enabled high-yield expression, efficient refolding without significant losses, and simplified purification—ultimately overcoming key challenges of bacterial systems such as low solubility while preserving functional integrity.

The pET-SUMO-GM plasmid construct enabled the efficient expression of the fusion protein, reaching 23–25% of total cellular protein, equivalent to approximately 100 mg per liter of bacterial culture. Compared to the published yield of GM-CSF obtained in a homologous eukaryotic production system using CHO cells (0.771 mg/L [16], 3.5 mg/L [17], 3 mg/L [18]), our yield is significantly higher. Considering the difference in the cost of culturing *E. coli* versus mammalian cells, we roughly estimate the production cost in the latter to be approximately 50-fold higher. It is important to note that this is a preliminary estimate and does not include the costs of isolation and purification. The SUMO tag not only enhanced solubility but also streamlined purification through Ni-IMAC chromatography, followed by precise cleavage with Ulp protease. Additional polishing steps, including hydrophobic chromatography, yielded rhGM-CSF with >99.5% purity, meeting stringent therapeutic standards. Importantly, the final product exhibited biological activity identical to the WHO international standard, as confirmed by in vitro TF-1 cell proliferation assays (ED50: 0.045 ng/mL vs. 0.043 ng/mL). In vivo validation in a cyclophosphamide-induced myelosuppression model further demonstrated its therapeutic efficacy, with a 3.4-fold increase in segmented neutrophils, highlighting its potential for clinical applications.

A critical advantage of our method is its ability to produce functional, non-glycosylated GM-CSF, confirming that glycosylation is not required for activity. While eukaryotic systems are often preferred for post-translational modifications, our data align with prior studies showing that bacterial-derived GM-CSF retains full receptor-binding and hematopoietic capabilities. The stepwise refolding protocol effectively restored the native conformation of the protein, addressing a major bottleneck in bacterial expression systems.

Looking ahead, this platform could be further optimized through strain engineering (e.g., *E. coli* SHuffle for improved disulfide bond formation) or alternative solubility-enhancing tags (e.g., MBP). Additionally, preclinical studies in neurodegenerative models could validate its neuroprotective potential, expanding its applications beyond oncology.

## 5. Conclusions

This study demonstrates an efficient *E. coli*-based platform for producing biologically active recombinant human GM-CSF using SUMO fusion technology, overcoming key challenges of prokaryotic expression systems. By leveraging the SUMO tag, we achieved high-yield expression (23–25% of total cellular protein) and developed an optimized purification strategy involving nickel affinity chromatography, Ulp protease cleavage, and hydrophobic chromatography, resulting in >99.5% pure rhGM-CSF. Despite initial inclusion body formation, the refolded protein exhibited bioactivity equivalent to the WHO international standard (ED50 0.045 ng/mL in TF-1 cell assays) and significantly restored neutrophil counts (3.4-fold increase, *p* ≤ 0.05) in a murine myelosuppression model, confirming its therapeutic potential. These results challenge the necessity of glycosylation for GM-CSF functionality and provide a cost-effective alternative to eukaryotic production systems. Future work could further optimize the platform through engineered *E. coli* strains for disulfide bond formation or evaluate neuroprotective applications in neurodegenerative disease models. The proposed method offers a scalable, economically viable solution for GM-CSF production, with broad implications for clinical and research use in oncology, hematology, and neurology.

## Figures and Tables

**Figure 1 cimb-47-00681-f001:**
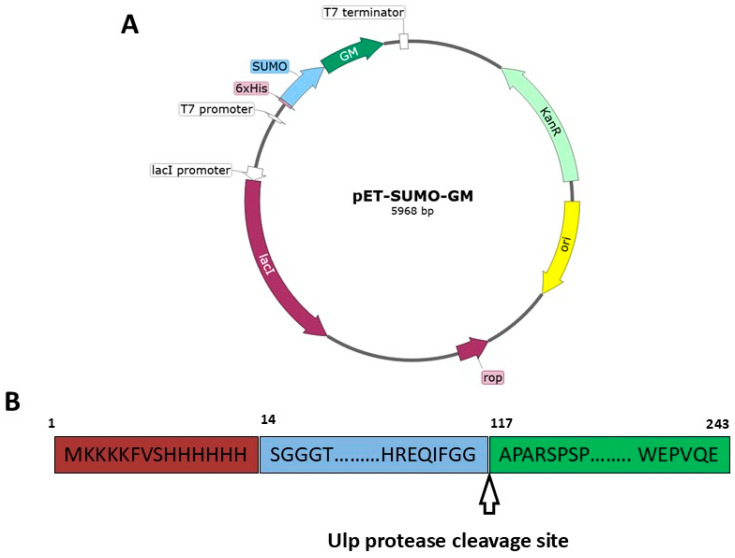
(**A**) Plasmid map of pET-SUMO-GM. (**B**) Schematic of the SUMO-GM fusion protein.

**Figure 2 cimb-47-00681-f002:**
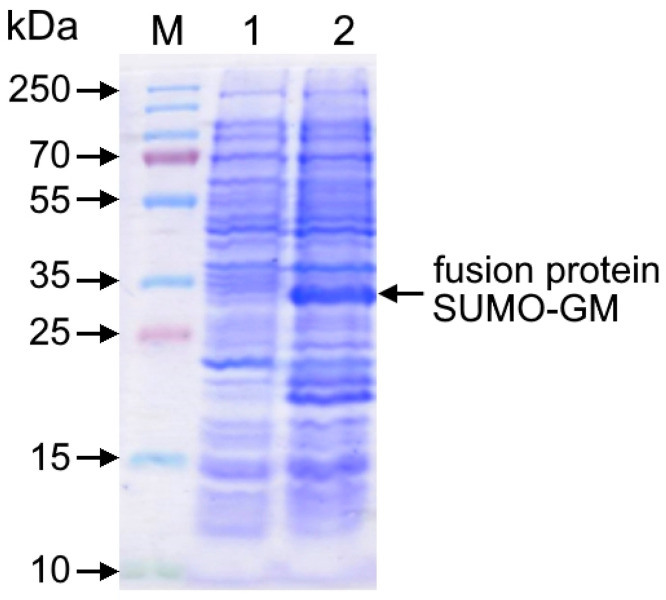
Electropherogram of cell lysates from the producer strain *E. coli* BL21(DE3)/pET-SUMO-GM. Electrophoresis was performed in 12% polyacrylamide gel (PAGE) with Coomassie R-250 staining. Lanes: M—protein molecular weight marker (10–250 kDa); 1—cell lysate before induction; 2—cell lysate 5 h after IPTG induction.

**Figure 3 cimb-47-00681-f003:**
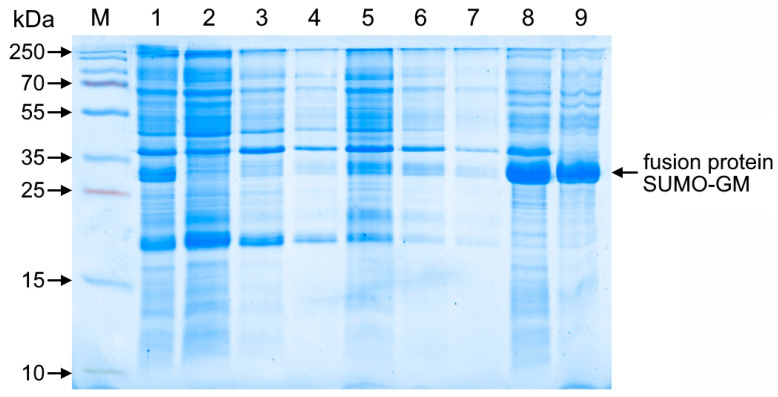
Electropherogram of SUMO-GM fusion protein samples after extraction and refolding. Separation was performed by 15% PAGE under non-reducing conditions with Coomassie R-250 staining. Lanes: M—molecular weight marker (10–250 kDa); 1—lysed cell suspension; 2—supernatant of lysed cell suspension; 3–4—wash with buffer (20 mM Tris-HCl, 5 mM EDTA, pH 8.0); 5–6—wash with buffer (20 mM Tris-HCl, 0.5% Tergitol 15-S-9, pH 8.0); 7—wash (20 mM Tris-HCl, pH 8.0); 8—denatured protein solution before centrifugation; 9—supernatant of denatured protein solution.

**Figure 4 cimb-47-00681-f004:**
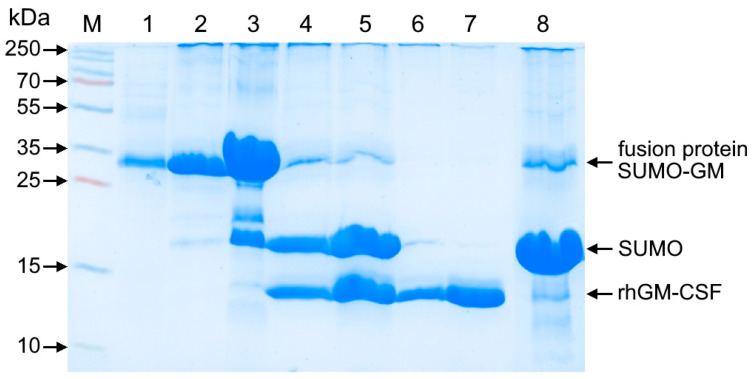
SDS-PAGE analysis of affinity-purified His-tagged SUMO-GM fusion protein after SUMO tag cleavage and removal. Separation was performed using 15% polyacrylamide gel under non-reducing conditions with Coomassie R-250 staining. Lanes: M—molecular weight marker (10–250 kDa); 1—refolded protein solution; 2–3—elution fractions of SUMO-GM fusion protein from first IMAC (fractions 1–2); 4–5—Ulp protease digest (1:50 enzyme/substrate ratio) of SUMO-GM (fractions 1–2); 6–7—flow-through fractions containing rhGM-CSF from second IMAC (fractions 1–2); 8—SUMO-containing eluate from second IMAC.

**Figure 5 cimb-47-00681-f005:**
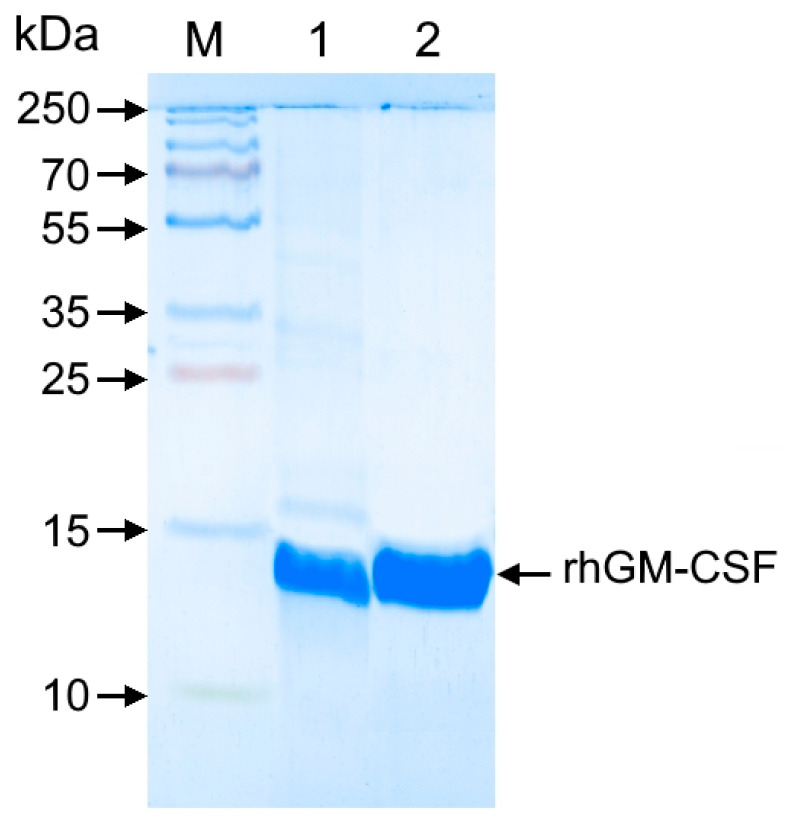
SDS-PAGE analysis of rhGM-CSF purification by hydrophobic chromatography. Proteins were separated on a 15% polyacrylamide gel under non-reducing conditions and visualized by Coomassie R-250 staining. Lanes: M—molecular weight marker (10–250 kDa); 1—pooled rhGM-CSF fractions after affinity chromatography; 2—rhGM-CSF fraction following HIC purification.

**Figure 6 cimb-47-00681-f006:**
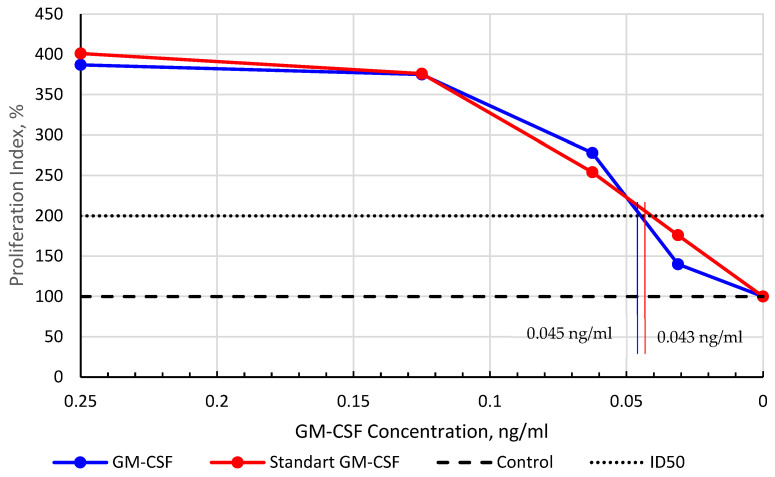
Dose–response curve of rhGM-CSF proliferative activity in TF-1 cells. The proliferation index was calculated as the ratio of sample optical density to control optical density, expressed as a percentage. The dashed line at 100% represents baseline proliferation in GM-CSF-free controls. ED50 denotes the concentration yielding two-fold greater proliferation than controls.

**Table 1 cimb-47-00681-t001:** Oligonucleotides used for plasmid construction.

Primer	Sequence (5′ → 3′)
GSUMO-F	tgagcggataacaattcccctc
SGM-R	ggctacgtgccggtgcgccaccaatctgctcacg
GM-SUMO-R	tggtggtgctcgagttattattcttgaacaggttcccaacaatc
SGM-F	cgtgagcagattggtggcgcaccggcacgtagcc

**Table 2 cimb-47-00681-t002:** Peripheral blood parameters in CBA mice following quadruple administration of rhGM-CSF (90 μg/kg, subcutaneous) against cyclophosphamide-induced myelosuppression.

Group	Leukocytes, 109/л	Segmented Neutrophils, 109/л	Leukocyte Differential, %				
			Eosinophils	Neutrophils		Monocytes	Lymphocytes
				Band Neutrophils	Segmented Neutrophils		
Control: Physiological saline	1.9 ± 0.2	0.10 ± 0.02	0.0 ± 0.0	6.3 ± 1.9	5.2 ± 1.0	11.2 ± 1.8	77.0 ± 4.7
Test: rhGM-CSF	2.2 ± 0.3	0.34 ± 0.06 * *p* = 0.008	0.0 ± 0.0	6.5 ± 1.6	15.3 ± 1.6 * *p* = 0.005	16.0 ± 1.7	62.2 ± 3.5 * *p* = 0.045
Intact	7.3 ± 1.0	1.65 ± 0.18	0.4 ± 0.2	2.2 ± 0.2	25.2 ± 5.7	5.4 ± 1.0	66.8 ± 5.8

Data are presented as mean ± standard error of the mean (SEM). Significant differences versus saline control (* *p* ≤ 0.05).

## Data Availability

The original contributions presented in the study are included in the article; further inquiries can be directed to the corresponding author.

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
