# Peer review of "Development of a Method for Producing Recombinant Human Granulocyte-Macrophage Colony-Stimulating Factor Using Fusion Protein Technology"

_cimb, 2025, doi:10.3390/cimb47090681_

Round 1
Reviewer 1 Report
Comments and Suggestions for Authors
This manuscript, titled “Development of a Method for Producing Recombinant Human Granulocyte-Macrophage Colony-Stimulating Factor Using Fusion Protein Technology”, presents a method for producing of pure biologically active recombinant Granulocyte-macrophage colony-stimulating factor (GM-CSF) using a cost-effective E. coli expression system, followed by refolding processes from inclusion bodies. Now GM-CSF has been approved in many countries for treatment of neutropenia following chemotherapy or transplantation, for graft failure, and for peripheral blood stem cell mobilization. (GM-CSF general description, www.ncbi.nlm.nih.gov/books/NBK13400/). However, most commercial GM-CSF preparations currently rely on expensive eukaryotic systems. The proposed method is notable because it yields biologically active monomeric GM-CSF protein identical to the standard preparation by refolding E. coli inclusion bodies. Inclusion bodies typically indicate high expression levels and facilitate protein purification but pose challenges related to protein refolding.
The manuscript is well-written and well-structured, encompassing a substantial amount of experimental work and providing valuable information that will undoubtedly interest researchers in molecular and cell biology as well as drug discovery.
However, the manuscript has some shortcomings. The introduction lacks detailed information on the structure of the native GM-CSF protein, including glycosylation and disulfide bonds, as well as an analysis of why eukaryotic production systems were initially preferred for commercial preparations. It is known that removal of the carbohydrate moiety—either through bacterial synthesis or mutagenesis—can actually increase the specific activity of GM-CSF (www.ncbi.nlm.nih.gov/books/NBK13400/).
Unfortunately, the manuscript is not free from some shortcomings. The introduction clearly lacks information on the structure of the natural GM-CSF protein itself (glycosylation, disulfide bonds), as well as an analysis of the reasons why eukaryotic production systems were first used for commercial GM-CSF preparations. Although it is known that the removal of removal of the carbohydrate moiety, either by synthesis in bacteria or mutagenesis, actually increases the specific activity of GM-CSF (www.ncbi.nlm.nih.gov/books/NBK13400/).
Minor points:
- Line 39 – A comma is missing between the keywords “protein chromatography” and “hematopoietic activity.”
- Line 142 – The sentence “to remove imidazole and sodium chloride from the protein solution” requires clarification regarding the source of sodium chloride, as it was not mentioned in the previously listed solutions.
- Line 248 – The figure caption describes the wash buffer composition for inclusion bodies as containing Triton X-100, whereas the Materials and Methods section lists the detergent as the milder Tergitol. Please clarify which detergent was used.
Author Response
- However, the manuscript has some shortcomings. The introduction lacks detailed information on the structure of the native GM-CSF protein, including glycosylation and disulfide bonds, as well as an analysis of why eukaryotic production systems were initially preferred for commercial preparations. It is known that removal of the carbohydrate moiety—either through bacterial synthesis or mutagenesis—can actually increase the specific activity of GM-CSF (www.ncbi.nlm.nih.gov/books/NBK13400/).
Thank you very much for this comment. We have included in the introduction section detailed information on the structure of GM-CSF and the reasons why the eukaryotic system is preferable.
Minor points:
- Line 39 – A comma is missing between the keywords “protein chromatography” and “hematopoietic activity.”
Thank you. We have fixed this error.
- Line 142 – The sentence “to remove imidazole and sodium chloride from the protein solution” requires clarification regarding the source of sodium chloride, as it was not mentioned in the previously listed solutions.
Thank you. We have fixed this error.
- Line 248 – The figure caption describes the wash buffer composition for inclusion bodies as containing Triton X-100, whereas the Materials and Methods section lists the detergent as the milder Tergitol. Please clarify which detergent was used.
Thank you. We have corrected this error. Tergitrol was used.

Reviewer 2 Report
Comments and Suggestions for Authors
This manuscript describes the development of a cost-effective Escherichia coli-based platform for producing biologically active recombinant human GM-CSF (rhGM-CSF) using SUMO fusion technology. The authors engineered the pET-SUMO-GM plasmid to express a 33 kDa fusion protein, which constituted 23–25% of total cellular protein but accumulated primarily in inclusion bodies. A multi-step purification strategy—nickel affinity chromatography, Ulp protease cleavage, and hydrophobic interaction chromatography—yielded >99.5% pure rhGM-CSF. Functional validation demonstrated equivalent in vitro activity to the WHO international standard and significant in vivo efficacy in a cyclophosphamide-induced murine myelosuppression model. The study concludes that prokaryotic systems can overcome solubility/folding challenges to produce therapeutic-grade rhGM-CSF, enabling broader clinical applications. However, several concerns should be addressed.
Major Concern:
The central objective of this study is to try to solve “the high cost of eukaryotic production systems.” However, the manuscript fails to provide any quantitative data comparing:
1). Production costs (per mg) of their SUMO-fusion E. coli system vs. eukaryotic systems (e.g., CHO cells).
2). Yield efficiency (mg/L) relative to established platforms. These metrics are essential to substantiate the cost-effectiveness claim and to contextualize performance against current industry standards.
Minor Concern:
- The lane labels in Figure 2 (Lanes: 1, 2, 3) do not match the descriptions provided in the figure caption. This mismatch can confuse readers about which sample or condition each lane represents.
- In Figure 4, Lane 8, why is the amount of SUMO much higher than rhGM-CSF and the fusion protein SUMO-GM? Specifically, based on the band intensities observed in Lane 8 of Figure 4, the detected level of free SUMO appears to exceed that of rhGM-CSF and the SUMO-GM fusion.
- Add a table comparing: a). Yield (mg/L) of their final product vs. published eukaryotic/prokaryotic systems; b). Estimated cost per mg.
- SUMO was chosen for "solubility enhancement," but no data proves it outperforms cheaper alternatives (e.g., MBP tags, codon optimization, or chaperone co-expression).
- Minor syntax errors (e.g., "stepwise dilution of the denaturing buffer" → "stepwise dilution in buffer").
Author Response
Major Concern:
The central objective of this study is to try to solve “the high cost of eukaryotic production systems.” However, the manuscript fails to provide any quantitative data comparing:
1). Production costs (per mg) of their SUMO-fusion E. coli system vs. eukaryotic systems (e.g., CHO cells).
2). Yield efficiency (mg/L) relative to established platforms. These metrics are essential to substantiate the cost-effectiveness claim and to contextualize performance against current industry standards.
We thank the reviewer for this valuable comment. Accordingly, we have revised the Discussion section to include data on the protein yield per liter of culture and a comparative cost analysis of the protein obtained in different expression systems.
The pET-SUMO-GM plasmid construct enabled the efficient expression of the fusion protein, reaching 23–25% of total cellular protein, equivalent to approximately 100 mg per liter of bacterial culture. Compared to the published yield of GM-CSF obtained in a homologous eukaryotic production system using CHO cells (0.771 mg/L [16], 3.5 mg/L [17], 3 mg/L [18]), our yield is significantly higher. Considering the difference in the cost of culturing E. coli versus mammalian cells, we roughly estimate the production cost in the latter to be approximately 50-fold higher. It is important to note that this is a preliminary estimate and does not include the costs of isolation and purification.
Minor Concern:
- The lane labels in Figure 2 (Lanes: 1, 2, 3) do not match the descriptions provided in the figure caption. This mismatch can confuse readers about which sample or condition each lane represents.
Thank you very much for your comment, it was a technical error, we have corrected it.
- In Figure 4, Lane 8, why is the amount of SUMO much higher than rhGM-CSF and the fusion protein SUMO-GM? Specifically, based on the band intensities observed in Lane 8 of Figure 4, the detected level of free SUMO appears to exceed that of rhGM-CSF and the SUMO-GM fusion.
Thank you for your question. The fact is that the fraction of the protein preparation that remains after the stage of proteolytic cleavage and elution of GM-CSF was applied to lane 8. This fraction should contain only SUMO, but in practice it contains trace amounts of the fused protein and GM-CSF.
- Add a table comparing: a). Yield (mg/L) of their final product vs. published eukaryotic/prokaryotic systems; b). Estimated cost per mg.
This is a valuable comment and in response to it we have added some information to the discussion section, but we ask permission from the reviewer not to include such a table in the text of the article. Our motivation is that the journal for which we prepared the article has a more molecular biology focus, while the indicators that the reviewer pays attention to are more biotechnological.
- SUMO was chosen for "solubility enhancement," but no data proves it outperforms cheaper alternatives (e.g., MBP tags, codon optimization, or chaperone co-expression).
This is a very good comment. Indeed, there are other approaches that allow us to regulate the yield and solubility of protein during synthesis in E. coli. Similar work has been carried out by our research group as well. Another article on these results is currently being prepared. Therefore, we considered it unnecessary to include these results in the current article in order to focus attention on the most successful of the designs we have developed.
- Minor syntax errors (e.g., "stepwise dilution of the denaturing buffer" → "stepwise dilution in buffer").
Thank you very much for your comment, we have changed the wording of this sentence to make it more understandable. Also, a comprehensive work was carried out to revise the text for errors.
